# Phototrophic purple bacteria as optoacoustic in vivo reporters of macrophage activity

Lena Peters[1], Ina Weidenfeld[2], Uwe Klemm[2], Anita Loeschcke[1], Robin Weihmann[1], Karl-Erich Jaeger[1,3], Thomas Drepper[1], Vasilis Ntziachristos[2,4] & Andre C. Stiel [2]

The morphology, physiology and immunology, of solid tumors exhibit spatial heterogeneity which complicates our understanding of cancer progression and therapy response. Understanding spatial heterogeneity necessitates high resolution in vivo imaging of anatomical and pathophysiological tumor information. We introduce *Rhodobacter* as bacterial reporter for multispectral optoacoustic (photoacoustic) tomography (MSOT). We show that endogenous bacteriochlorophyll *a* in *Rhodobacter* gives rise to strong optoacoustic signals >800 nm away from interfering endogenous absorbers. Importantly, our results suggest that changes in the spectral signature of *Rhodobacter* which depend on macrophage activity inside the tumor can be used to reveal heterogeneity of the tumor microenvironment. Employing non-invasive high resolution MSOT in longitudinal studies we show spatio-temporal changes of *Rhodobacter* spectral profiles in mice bearing 4T1 and CT26.WT tumor models. Accessibility of *Rhodobacter* to genetic modification and thus to sensory and therapeutic functions suggests potential for a theranostic platform organism.

[1] Institute of Molecular Enzyme Technology (IMET), Heinrich Heine University Düsseldorf, Forschungszentrum Jülich GmbH, Jülich 52425, Germany. [2] Institute of Biological and Medical Imaging (IBMI), Helmholtz Zentrum München, Neuherberg 85764, Germany. [3] Institute of Bio- and Geosciences (IBG-1): Biotechnology, Forschungszentrum Jülich GmbH, Jülich 52425, Germany. [4] Chair of Biological Imaging and Center for Translational Cancer Research (TranslaTUM), Technische Universität München, München 81675, Germany. These authors contributed equally: Lena Peters, Ina Weidenfeld, Uwe Klemm. Correspondence and requests for materials should be addressed to T.D. (email: t.drepper@fz-juelich.de) or to A.C.S. (email: andre.stiel@helmholtz-muenchen.de)

Solid tumors are highly heterogeneous, containing sub-populations of genetically—and phenotypically distinct cells[1]. These microenvironments are characterized by spatial differences in oxygen tension, pH, nutrient availability, and immune system accessibility. In addition, tumor microenvironments exhibit a variable distribution of specific cells strongly implicated in tumor transition towards malignancy such as tumor-associated macrophages (TAM)[2,3]. This heterogeneity further complicates our understanding of tumor biology and disease progression, and challenges therapeutic interventions[4]. In vivo high-resolution imaging has been a fundamental tool for spatially resolving tumor morphology, physiology, or biochemical composition, thereby allowing to unravel underlying driving forces of tumor biology. Intravital microscopy of the tumor microenvironment is broadly employed in oncological research but suffers from a limited field-of-view and penetration depth[5]. Radiological methods, such as positron emission tomography (PET) can image tumor pathophysiology at much larger scales but exhibit limited spatial resolution[6]. Magnetic resonance imaging (MRI)[7], X-ray, computer tomography (CT), or ultrasonography enable high-resolution visualization of morphology and functional tumor parameters, but detailed sensing of pathophysiological parameters over time is challenging due to the limited sensitivity afforded. Moreover, techniques such as PET or MRI require large infrastructure out of the reach of many research institutions.

Multispectral optoacoustic (OA) tomography (MSOT) combines optical contrast with ultrasound resolution enabling high-resolution real time in vivo imaging well-beyond the 1 mm penetration depth typical of microscopy methods[8,9]. Therefore, it is emerging as a particularly interesting alternative imaging method in cancer research. However, in label-free mode, it only records a limited number of factors of tumor pathophysiology, e.g., angiogenesis. Therefore, several agents have been considered for extending the optoacoustic capacity, including nanoparticles and targeted chromophores which however are only transient and do not allow longitudinal studies (reviewed in ref. [10]). Transgenic expression of labels has also been considered for OA imaging, in particular fluorescent proteins as well as the pigments melanin, violacein, or an indigo dye produced by enzymatic cleavage of X-gal (reviewed in ref. [11]). Major challenges of these genetically encoded labels are the absorbance in the visible part of the spectrum hampering their detection deep in tissues due to strong absorbance of blood at those wavelengths. In contrast, the absorption spectrum of melanin extends to the near-infrared region (NIR) however lacks distinct peaks[12] making it difficult to separate its OA signal from background noise. Moreover, synthesis of melanin in mammalian cell lines for longitudinal studies is often precluded by long-term toxic effects[13].

Here, we propose an alternative OA reporter based on a bacterial system. Bacteria have been considered for visualization[14] together with therapeutic purposes[15,16] like release of anti-tumorigenic payloads, acting as a vector for delivering transgenes into mammalian recipient cells[17] or directly hampering tumor cell proliferation[18,19]. However, no study has attempted to use bacteria for in vivo monitoring of pathophysiological processes. Here, we consider facultative phototrophic purple bacteria that are intrinsically rich in bacteriochlorophyll $a$ (BChl $a$), which exhibits several favorable characteristics as OA labels: two clearly defined spectral bands in the near-infrared range ($\lambda_{max}$ of 800 and ~860 nm) with high molar absorptivity[20,21] which are red-shifted compared to all genetically encoded labels reported for OA so far, suggesting a unique potential for clear differentiation of bacterial signal from endogenous absorbers prevalent in tissue such as blood hemoglobin and lipids. Compared to free BChl $a$ that absorbs primarily at ~770 nm in solution, the peak signature in phototrophic bacteria is heavily shaped by the BChl $a$ being embedded in the membrane-bound photosynthetic machinery. This spectral tunability can basically be applied for reporter approaches. Since a recent work has reported the targeting of *Rhodobacter sphaeroides* to various solid tumors[22], purple non-sulfur bacteria of the genus *Rhodobacter*, which have been extensively used as model organisms to study the regulation and function of anoxygenic photosynthesis[23–25], may be particularly suited for developing MSOT compatible reporters applicable for in vivo tumor studies.

However, different factors and microenvironmental conditions are likely to influence the fate of *Rhodobacter* and hence the BChl $a$-derived OA signal. Those are, for example, the initial pigment levels before injection which can be affected by individual strain characteristics and cultivation conditions, as well as components of the bacterial cell envelope including lipopolysaccharides (LPS) and capsular or exopolysaccharides (EPS) that might further affect the interaction between the bacterial cell and the microenvironment inside the tumor. Therefore, we investigate the suitability of different *Rhodobacter* species for optoacoustic signal generation, explore their fate after intratumoral injection and the potential of their distinct spectral signature to carry any additional information from the tumor microenvironment.

## Results

**BChl $a$ production in different purple bacterial strains.** Bacteria used for tumor imaging are required to tolerate the temperature of their respective host. Phototrophic purple non-sulfur bacteria, however, generally prefer growth temperatures of 30 °C and below. Furthermore, in these bacteria efficient photopigment synthesis, which is essential for sensitive OA detection, is generally favored by lowered oxygen accessibility and high nutrient availability. To identify the optimal strain/cultivation combination for MSOT applications, we first analyzed the viability, pigment production, and corresponding absorption properties of different photosynthetic bacteria at 37 °C under anaerobic and microaerobic conditions in various media and light regimes (see Methods section and legend of Supplementary Table 1 for details on growth conditions).

In addition to *Rba. sphaeroides* ATH 2.4.1, which could successfully be used for tumor targeting and visualization by detecting BChl $a$-dependent fluorescence[22], we analyzed the non-sulfur purple bacteria strains *Rba. capsulatus* SB1003 and B10S as well as the SB1003 mutant strain $\Delta crtJ$ carrying a deletion within the repressor gene involved in $O_2$-mediated inhibition of BChl $a$ biosynthesis[26]. Furthermore, we analyzed *Rhodospirillium centenum* ATCC 43720. In contrast to the representatives of the genus *Rhodobacter*, *Rsp. centenum* is less well-characterized but is capable of growing at elevated temperatures up to 42 °C[27].

Surprisingly, with exception of *Rba. capsulatus* B10S and *Rsp. centenum*, the chosen strains showed substantial growth at 37 °C under all tested anaerobic and microaerobic conditions. Phototrophic growth in the mixed medium (i.e., a composition of minimal and complex media) under anaerobic conditions and in the presence of infrared (IR) light illumination resulted in highest BChl $a$ absorption per cell (all results of the comparative cultivation assay are shown in Supplementary Table 1). Under these conditions, the wild-type SB1003 and $\Delta crtJ$ strains of *Rba. capsulatus* exhibited the highest BChl $a$ absorption with a ratio $OD_{860/660}$ of 2.15 to 2.48. In agreement with earlier studies[23–25,28] we observed that increased $O_2$ availability resulted in reduction of photopigment synthesis for all *Rhodobacter* strains. Thus, in contrast to *Rsp. centenum*, which consistently exhibited a low pigment-to-cell ratio under all tested cultivation conditions, the *Rba. capsulatus* strains SB1003 and $\Delta crtJ$ as well as *Rba.*

*sphaeroides* showed robust growth at 37 °C together with high BChl *a* production favorable for usage in OA experiments.

***Rba.* absorption spectra and in vitro OA characterization**. Next, we grew *Rba. capsulatus* SB1003, its Δ*crtJ* variant, and *Rba. sphaeroides* under the optimal conditions for photopigment formation, i.e. in the absence of oxygen, in mixed medium and with IR illumination at 37 °C, in order to achieve optimal signal intensities for subsequent OA analyses. In all cases, characteristic peaks of cellular BChl *a* were observed at around 800/850 nm (*Rba. sphaeroides*) and 800/860 nm (*Rba. capsulatus*) (Supplementary Figure 1A, for convenience we name the 850/860 nm peak exclusively 860 nm throughout the manuscript), and in vitro OA spectra of the bacteria recorded with MSOT were consistent with their corresponding optical absorption spectra (Supplementary Figure 1B). In general, the OA signal of a pigment normalized by its corresponding absorbance is a measure of the OA signal generation efficiency (photoacoustic signal generation efficiency, PGE). PGEs normalized to that of a reference solution of methylene blue, yielding 2.52 in the case of *Rba. sphaeroides*, 2.18 in the case of *Rba. capsulatus* SB1003 and 3.04 in the case of the Δ*crtJ* variant are comparable to the PGE of the well-established OA label melanin (3.36). In comparison to blood, the raw OA signal at 850 nm of ~2.6 x $10^6$ cells in the imaging volume (5 µl) equals that of fully oxygenated sheep's blood. This is one order of magnitude less cells than would be needed with melanin producing bacteria[12].

**OA characterization of *Rba.* in 4T1 mouse allografts**. Next, we examined the behavior and optoacoustic performance of these three strains after injection directly into 4T1 mouse mammary gland breast tumors grafted onto the backs of BALB/c nude mice. Our goal was to assess how good *Rhodobacter* can be detected in the tumor using OA and if the spectral features facilitate unmixing of the signal from endogenous absorbers in vivo providing a high signal to noise visualization. In addition, we examined whether bacterial proliferation could be optoacoustically resolved within the tumor. Therefore, bacteria ($1 \times 10^8$ cfu) grown under ideal conditions for BChl *a* production were injected into the tumors (n = 3 per strain, Supplementary Table 2 for information on all mice imaged in this study), followed by consecutive days of examination using MSOT. A control mouse was injected with phosphate-buffered saline (PBS, *n* = 1).

As a first step for in vivo application, we evaluated whether components of *Rhodobacter* cells could have a detrimental effect on tumor growth, precluding its use as a reporter system. None, of the bacterial strains significantly altered tumor growth or mouse body weight suggesting the bacteria exhibit no immediate anti-tumorigenic effects or systemic toxicity (Supplementary Figure 2). This observation could be further corroborated by the in vitro cell metabolic activity assay (MTT assay for colorimetric assessment of cellular NAD(P)H-dependent oxidoreductase activities). Here, heat-denatured *Rhodobacter* cells showed no negative effects on growth and viability of 4T1 cells (Supplementary Figure 3).

To identify *Rhodobacter*-specific signals, OA data were collected before and at different timepoints after injection into the tumor and subsequently reconstructed using a model-based algorithm. Averaged OA spectra of the tumor region clearly show the characteristic 800-nm and 860-nm peaks as the spectral signature of the injected bacteria, along with predominant spectral features reflecting tumor physiology, such as the slope and peak at 760 nm, which indicates the presence of deoxyhemoglobin (Supplementary Figure 4). Interestingly, in tumors, with all bacterial strains the two-peak signature of the averaged

OA spectra disappeared over time; in the two *Rba. capsulatus* strains, the peak at 860 nm was additionally blue-shifted to ~850 nm (Supplementary Figure 4A–I). To examine this phenomenon more closely, we unmixed the OA data now using an adaptive match filter and the 850-nm peak (*Rba. sphaeroides*) and 860-nm peaks (*Rba. capsulatus* SB1003 and Δ*crtJ*) as target spectra (see Methods section). This strategy allowed bacterial chlorophyll to be more faithfully detected even at later timepoints (Fig. 1a–c) with only a low fraction of possible false positives in the PBS control (Fig. 1e). Spectral analysis of only the unmixed pixels makes the abovementioned changes in the spectral signature even more apparent, pointing clearly to a temporal change in light absorption at 800 nm in the overall bacterial spectra (Fig. 1f–h). Furthermore, the unmixed results show relatively constant overall OA intensity over time with only a slight initial increase (Fig. 1a–d). The data suggest no strong growth of the bacteria inside the tumor. After the last timepoint the animals were sacrificed, tumors cryo-sectioned and preserved for histological analysis (see below).

**Analysis of the changed spectral signature**. The observed spectral shift of BChl *a*-mediated absorption suggests a change of photosystem architecture in *Rhodobacter* cells in response to processes that exclusively occurred after tumor injection; a spectral transition can be monitored in vivo by the applied MSOT approach. Therefore, the spectral shift could be employed to develop a reporting strategy suitable for analyzing dynamic processes in tumors. Consequently, this observation has led us to further explore the reasons for this spectral shift. First, we could rule out alterations in endogenous background absorbers because the spectra of the entire tumor in the PBS-injected control animal showed no substantial changes between 800 and 860 nm (Supplementary Figure 4O). Moreover, simulating convoluted spectra of the 800 and 860 nm signature spectra together with spectra of the major natural absorbers occurring in tumors (e.g., oxygenated and deoxygenated hemoglobin or lipids) suggest that the spectral change we observed for bacterial BChl *a* inside the tumor over time cannot result from such a convolution, although, tumor heterogeneity implies that such global spectral interpretations should be treated with caution (see Supplementary Note 1: Potential confounders of the spectral signature). Selective photobleaching of the 800-nm peak could also be excluded since continuously recorded OA spectra of *Rba. capsulatus* SB1003 in an in vitro phantom showed no changes in the peak ratio (Supplementary Figure 5). Furthermore, a second in vivo experiment, where MSOT was conducted only at the first and last timepoint to avoid induction of photobleaching effects via repeated BChl *a* excitation, showed the same spectral changes as observed for the initial experiment with repetitive MSOT measurements (Supplementary Figure 6). As a next step we studied whether the spectral changes could be due to metabolic changes in the bacteria resulting in a reconfiguration of the photosynthetic machinery[29]. To this end, we performed cultivation experiments, where *Rhodobacter* growth conditions were changed in an attempt to partially reflect conditions inside solid tumors (i.e., absence of light, presence of hydrogen peroxide, acidification, and availability of nutrients) to induce potential microbial processes that might be responsible for the tumor-induced shift of BChl *a* absorption. However, none of the tested conditions were able to induce similar spectral effects that have been observed in vivo (Supplementary Figure 7). In addition, we injected *Rba. capsulatus* SB1003 that have been inactivated by cell lysis (French press) into mice bearing 4T1 tumors (*n* = 3). We likewise observed a similar spectral change, however, at a much faster rate than utilizing intact cells (Supplementary Figure 8 for ROI spectra and

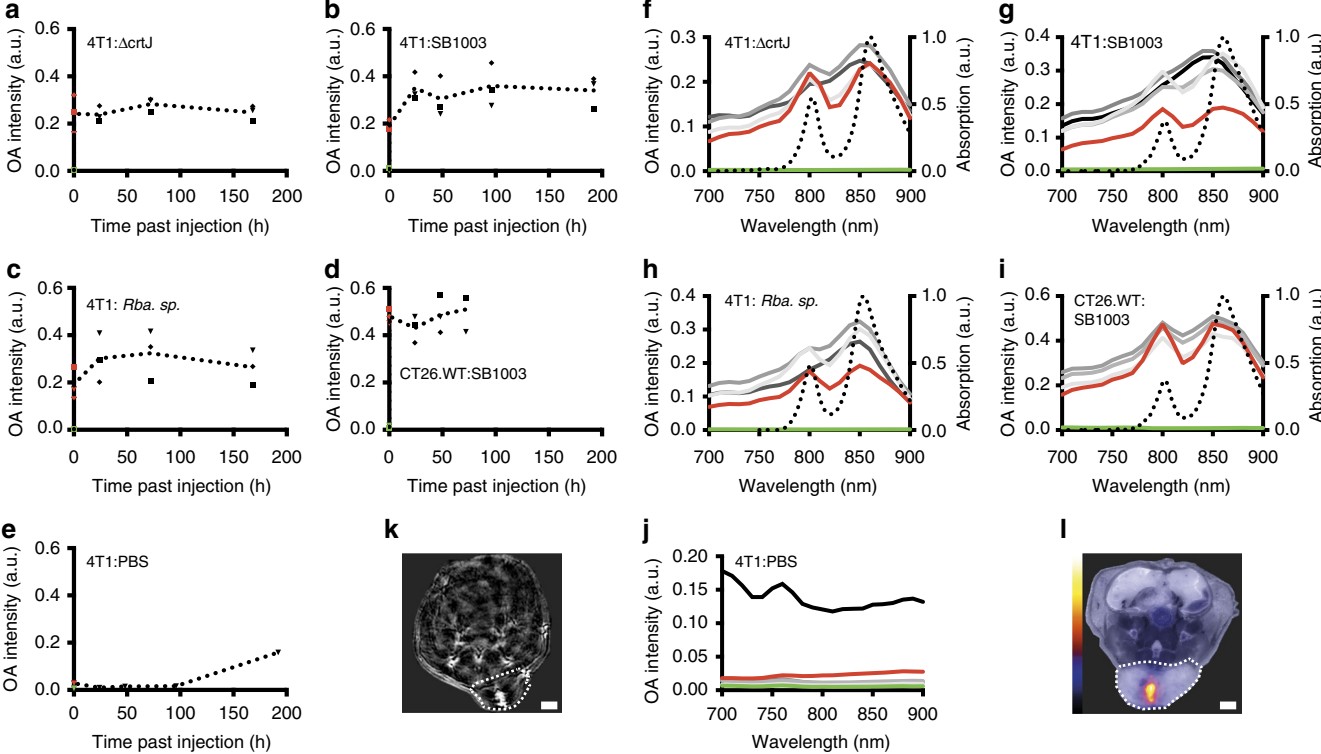

**Fig. 1** Characterization of the OA signal and spectra of phototrophic bacteria injected into tumors. **a–e** OA signal of phototrophic bacteria *Rba. capsulatus* SB1003 (SB1003), mutant Δ*crtJ* (Δ*crtJ*), and *Rba. sphaeroides* (*Rba.* sp.) injected into 4T1 and CT26.WT tumors. For each strain, data are shown from three mice before injection (green), immediately after injection (red) and at several timepoints after injection (black); dotted line represents the average (*n* = 3). The signal was calculated as the sum of the signal intensities at the maximum wavelength of all unmixed pixels in the region of interest summed over all slices. Target spectra for unmixing were the 860-nm absorption peaks of all strains. Data from PBS-injected mice used as a control. **f–j** Change of the OA spectra of the same mice shown in **a–e** over time. Shown are the average spectra of all three mice per strain and timepoint. Data from before injection are shown in green; immediately after in red and at increasing timepoints as shades of gray (24, 48, 72/96, 168/192 h). The spectrometrically determined absorption spectrum of the injected agent is shown for comparison as a dotted line. **k** MSOT slice of the mid-section of a 4T1 tumor-bearing mouse at 24 h after injection with *Rba. capsulatus* Δ*crtJ* and illuminated at 720 nm. **l** Cryosection of 4T1 tumor-bearing mouse at terminal timepoint (168 h) after injection with *Rba. capsulatus* SB1003. Excitation 740/40 nm bandpass, emission 850 nm longpass, exposure 10 s, gain 40. The analyzed region of interest (i.e., tumor) is indicated as a dotted line in both images (k, l). Scale bar: 2 mm

Supplementary Figure 14 for maximum intensity projections). This suggests that neither an adaptation process of live *Rhodobacter* cells in response to the tumor environment nor the process of cell lysis solely caused the observed spectral changes. Subsequently, we analyzed if degradation of the photosynthetic membrane—a process that might be initiated when *Rhodobacter* cells are inactivated by the immune system—could be causative for the spectral changes. Therefore, we first recorded the spectral signature of *Rba. capsulatus* SB1003 cell extracts where the photosystem was denatured by prolonged incubation at 95° in comparison to native (French press) extracts (Supplementary Figure 9). While the native cell extracts showed the 800 nm/860 nm signature as observed for intact cells indicating undamaged membranes and photosynthetic complexes, the heat-denatured samples displayed a distinct 770 nm absorbance indicative of free BChl *a*. Ultimately, we analyzed the possibility of phagocytotic uptake of *Rhodobacter* by macrophages which are not only commonly involved in host–pathogen defense, but are also prevalent in tumors. To this end we co-incubated *Rba. capsulatus* SB1003 cells together with Ana-1 macrophages or 4T1 tumor cells. Subsequently, the OA spectra of both the mammalian cells and their corresponding supernatants were recorded separately (Fig. 2 and Supplementary Figure 10). We could not observe any spectral information from 4T1 cells indicating no or only marginal uptake of *Rhodobacter*. In contrast, a strong

*Rhodobacter*-mediated MSOT signal was detected in Ana-1 macrophages 24 h and up to 144 h after onset of co-cultivation showing a spectral shift similar to the one observed in the tumor experiments and hence indicating that uptake of bacteria by macrophages is responsible for the spectral conversion. MSOT spectra of the cell culture media provide mirror images after 24 h with the classical two peaked spectral signature of live *Rhodobacter* remaining in the media of 4T1 cells while signal of free-living *Rhodobacter* cells is no longer detectable in media of macrophages. Indeed, microscopic analysis of the cells revealed that the bacteria are found inside Ana-1 cells (Fig. 2e). Regarding the intracellular fate of *Rhodobacter* taken up by macrophages we found signals of lysosomal marker lamp-1 and signals indicating ingested *Rhodobacter* but, surprisingly, observed no significant colocalization (Fig. 2f and Supplementary Figure 11).

In tissues, analysis of immunohistochemistry of tumors injected with *Rhodobacter* SB1003 utilizing anti-macrophage marker F4/80 shows a clear colocalization of macrophages and residual fluorescence signal in Cy7 channel from disintegrated *Rhodobacter* thereby confirming our in vitro results (Supplementary Figure 12). The region influenced by *Rhodobacter* injection furthermore includes tissue patches positive for cleaved caspase 3 indicating apoptosis in macrophages over time after uptake and disintegration of *Rhodobacter* cells. However, with macrophage viability decreasing only 10–15% within the first 24 h after

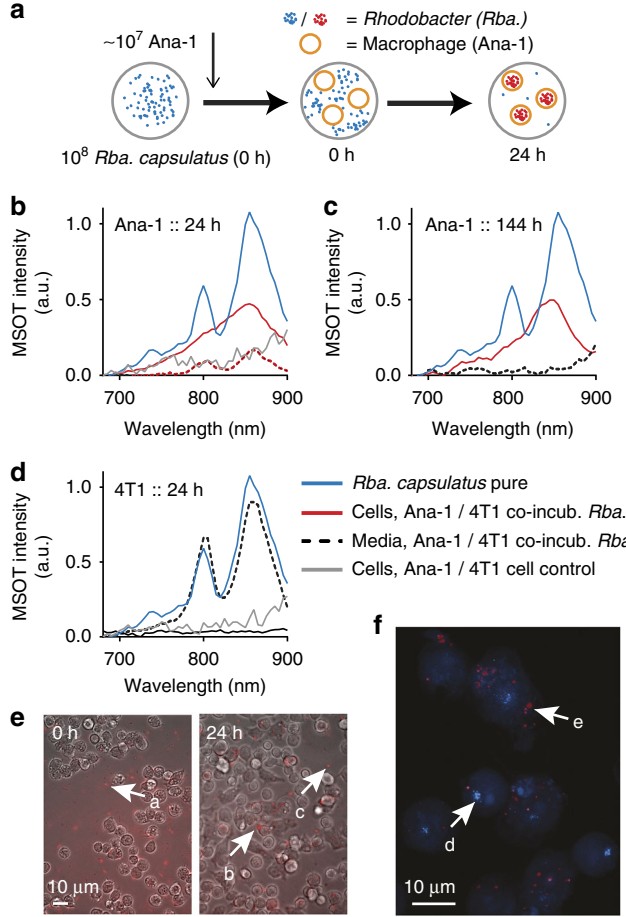

**Fig. 2** Addition of *Rba. capsulatus* SB1003 to Ana-1 macrophages and 4T1 tumor cells in vitro. Bacteria and mammalian cells were co-incubated at a ratio of 50:1, this being the maximum number of bacteria that can be taken up by one phagocyte. **a** Schematic of *Rhodobacter* getting ingested by macrophages resulting in a spectral shift as indicated by the blue and red coloration. **b**, **c** OA spectra of Ana-1 cells and supernatant after 24 and 144 h of co-incubation with *Rba. capsulatus* in comparison with the spectra of pure *Rhodobacter*. **d** 24 h data for 4T1 cells. **e** Representative images of Ana-1 macrophages in the presence of *Rhodobacter* at 0 and 24 h. After mixing bacterial and macrophage cells free swimming *Rhodobacter* (a) can be observed, while after 24 h the signal originates predominantly from inside the Ana-1 cells (b) with only a small number of bacteria identifiable outside (c). For 0 h fluorescence was recorded with a mCherry filter set, for 24 h with Cy7. Because of the high motility of *Rhodobacter* cells, fluorescence and brightfield images do not overlay. The full time course can be found in Supplementary Figure 10. **f** Fixated sample stained for lysosomal marker Lamp-1 (d, blue) and additionally recorded with Cy7 filter set for fluorescence of disintegrated *Rhodobacter* (e). Images for the individual channels as well as for the control, Ana-1 without addition of *Rba.*, can be found in Supplementary Figure 11. All scale bars 10 μm

*Rhodobacter* co-cultivation in vitro (Supplementary Figure 10) while a strong OA shift is detectable as shown in Fig. 2, we conclude that macrophage death is not a prerequisite for the disintegration of *Rhodobacter* leading to spectral changes.

Our results suggest so far that the temporal change in spectral signature we observed with intratumoral *Rhodobacter* reflects an uptake of the bacterial cells primarily by macrophages via phagocytosis. Likely, a subsequent disintegration of the bacteria in macrophages might be accompanied by a substantial release of BChl *a* molecules which could finally explain the spectral shift observed in vivo. However, since BChl *a* monomers in solution

exhibit an absorption maximum at ~770 nm the red shift may suggest a spontaneous aggregation of the photopigment upon macrophagal uptake. In principle, BChl *a* aggregation can lead to a red-shifted absorption as observed in vitro (e.g., in solvents with low ethanol concentrations as shown in Supplementary Figure 13). In previous in vitro studies, a similar spectral shift of BChl *a* is for instance described in *n*-octyl-β-D-glucopyranoside (β-OG)-based micellar systems ($\lambda_{max}$ ~850 nm[30],) or BChl *a* that has been covalently linked to 1-palmitoyl-2-hydroxy-*sn*-glycero-3-phosphatidylcholin ($\lambda_{max}$ = 824 nm[31]). The absorption spectra of BChl *a* aggregates thus resemble those recorded from *Rhodobacter* cells that have been internalized by macrophages suggesting induction of aggregate formation of BChl *a* monomers that might be formed in a micellar or membranous environment.

**Spatial analysis of OA spectra**. We next explored whether the changes in the spectral signature occurred uniformly throughout the tumor or were localized in certain regions. Analysis of spectra of pixels for which *Rhodobacter* signals were detected after unmixing showed that the two peaks at 800 and 860 nm were visible in the complete tumor immediately after injection. However, over time, the 800-nm peak selectively and gradually disappeared in the tumor edge region with the two-peak signature persisting only in the tumor core (exemplary images Fig. 3 and images of all studied mice at all timepoints Supplementary Figure 14).

In a next step we elucidated whether the characteristic spatiotemporal changes can reflect differences between tumor models. We therefore injected *Rba. capsulatus* SB1003 cells into CT26.WT tumors of accordingly xenografted mice ($n = 3$), recorded and processed data as described above. Comparison of pixel derived spectra at equivalent timepoints derived from 4T1- and CT26.WT mice harboring *Rba. capsulatus* SB1003 cells show that the change in the spectral signature is less pronounced in CT26.WT mice, which might reflect a lower density of tumor-associated macrophages[32] (Figs. 1d, i, 3b as well as Supplementary Figure 14). Our work thus shows that the spectral signature which is most probably linked to the uptake and disintegration of *Rhodobacter* by macrophages likely reflects their number, accessibility and activity in the tumor microenvironment.

**Spatial analysis of OA spectra in non-tumor tissue**. The uptake and disintegration of *Rhodobacter* cells in tumor tissue is a slow process which occurs, depending on the localization, on the scale of days. We were interested whether in healthy tissue this process occurs on different timescales. To this end, we injected strongly vascularized muscle tissue with $1 \times 10^8$ cfu *Rba. capsulatus* SB1003 ($n = 2$). Consistent with the expected uptake by blood circulating macrophages the spectral changes occurred faster (Fig. 3c as well as Supplementary Figure 14). Thus, the unique spectral shift *Rhodobacter* shows upon internalization by macrophages can also be used as a marker of macrophage presence and activity in other tissues, e.g. in the context of studying infection.

**Discussion**

We could demonstrate that BChl *a* accumulating purple bacteria can serve as label for OA imaging exhibiting outstanding spectral characteristics. In addition, we provide preliminary evidence that the characteristic two-peak BChl *a* spectra together with the spatiotemporal changes of the spectral shape depend on macrophage uptake and action. These properties can be used, e.g. in the case of tumors, to report macrophage accessibility in the tumor microenvironments. The strength of the BChl *a* OA signal (per unit of absorption) is comparable to that of the well-characterized strong absorber melanin. Moreover, the two BChl *a* peaks at 800 and 860 nm provide an excellent target spectrum for unmixing

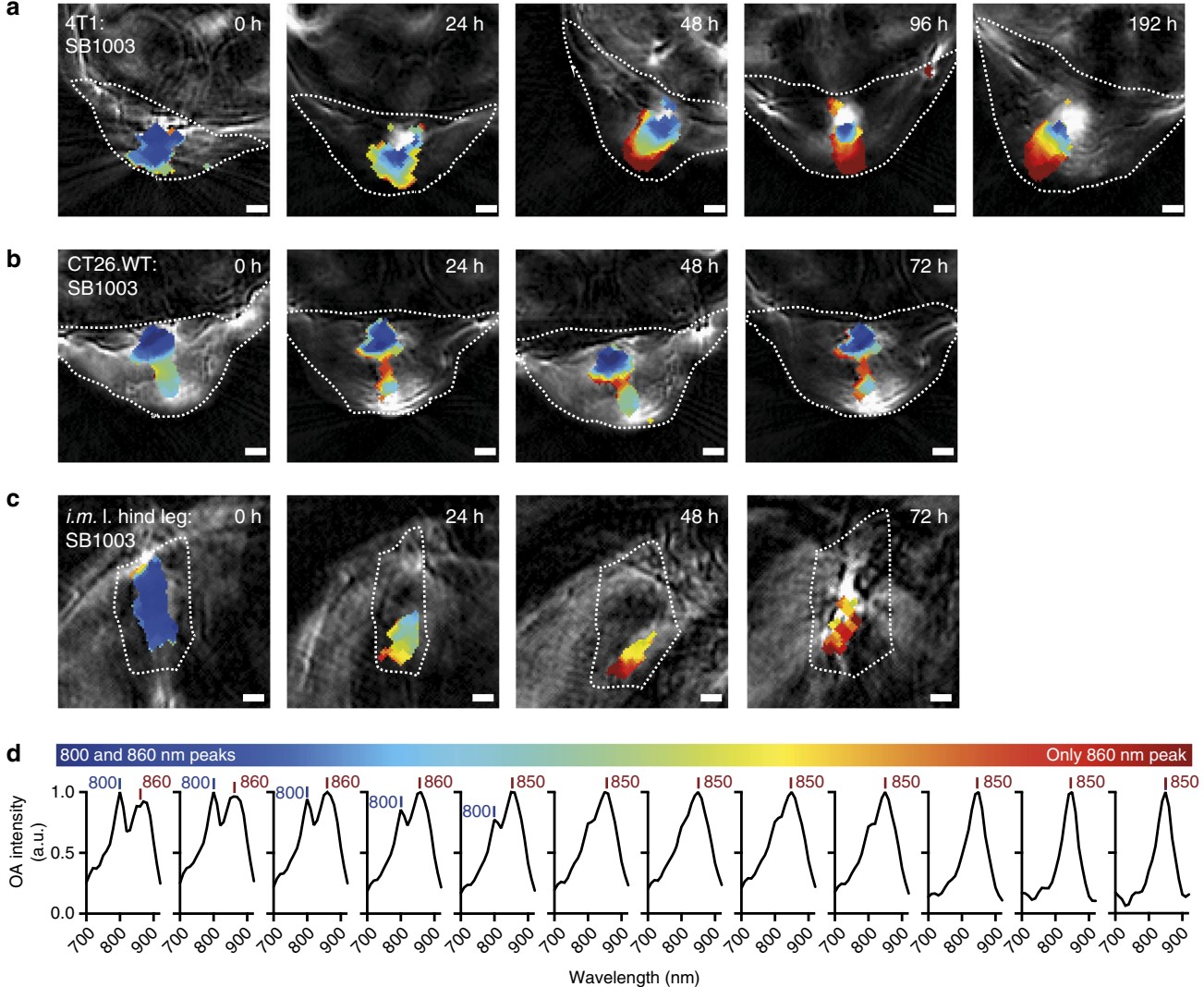

**Fig. 3** Spatiotemporal change of the bacterial spectral signature in the tumor environment and the muscle. A 4T1 (**a**) and a CT26.WT (**b**) tumor-bearing mouse intratumorally injected with Rba. capsulatus SB1003 is shown together with the hind leg muscle with the same injection (**c**). Shown are representative reconstructed images overlaid by unmixed pixels in false colors. Color coded regions indicate a mixture of 800 nm/860 nm peaks or predominance of the 860-nm peak as visualized in **d**. Shown are mean spectra of all regions of the same color. Unmixed pixels are shown as black if automatic peak assignment failed. Strong signal (white) in the reconstructed images at 720 nm relates to high absorption by blood or other absorber that has not been identified by the unmixing procedure. Data for all mice at all timepoints can be found in Supplementary Figure 14 (**a** $n = 3$, **b** $n = 3$, **c** $n = 2$). The analyzed region of interest (i.e., tumor) is indicated as a dotted line. All scale bars, 1 mm

and allow the study of regional effects inside the tumor with high resolution. Remarkably, we found reproducible ($n = 3$) spatiotemporal variation of the spectral pattern depending on the strain, the tumor model and the localization within the tumor at different times after injection. We believe that the basis for this shift is the phagocytosis and disintegration of the bacteria by macrophages and subsequent aggregation of the released BChl $a$. Our results clearly link the change of the spectral signature to macrophage activity in different regions of the tissue, which is an important contribution to understanding the tumor microenvironment and the spatially and temporally diverse plasticity, activity and overall functionality of macrophages[33], especially, since M2-like tumor-associated macrophages (TAM) are critical for shaping this microenvironment[34]. Moreover, the functionality of macrophages in respect to their phagocytic activity is linked to the situation in the tumor or to therapeutic intervention—e.g., data suggest that Bevacizumab might inhibit phagocytotic activity of TAMs[35]. Presently, detailed in vivo insight into macrophage

function has been obtained by intravital microscopy (IVM), however, the limited field-of-view of IVM precludes a comprehensive view on the whole tumor elucidating macrophage activity on the level of the tumor microenvironment. Whole animal imaging of macrophages inside the tumor has been shown with optoacoustics before[36] lacking however the visualization of activity which is provided by using *Rhodobacter* as a reporter agent. The present work therefore provides a starting point for applying OA imaging and *Rhodobacter* strains as reporters for studying the pathophysiological role of macrophages in preclinical cancer research and can help to develop strategies for bacterial tumor therapy.

Our results further show a dependence of the spatiotemporal pattern on the employed *Rhodobacter* strain suggesting an effect of variable cell envelope components such as EPS on the time course of bacterial cell lysis inside macrophages. EPS are constituents of the bacterial capsule layer surrounding most strains of *Rba. capsulatus*[37]. In initial studies, however, it could

be demonstrated that the composition and relative amount of the capsule seem to be strain-specific[38], and that the formation of capsule polysaccharides in *Rhodobacter* depends on the cultivation conditions, growth phase and the intrinsic quorum sensing system[39]. However, in contrast to free-living bacteria such as *Rhodobacter*, where EPS is mainly involved in protecting the cells against environmental threats, bacterial capsules can also have an important role as virulence factors during the infection process of human pathogens. On the other hand, EPS can also negatively affect intramacrophage survival and cell division as, for example, recently reported for *Salmonella*[40]. Therefore, the here described reporter properties of different *Rhodobacter* strains might also be considered as a sensor tool to elucidate the effect of different cell envelope components on the immune response and/or survival inside phagocytic cells using MSOT.

Finally, the identification of macrophages as promising therapeutic targets for chronical inflammatory and metastatic diseases further underpins the requirement for bacterial agents allowing to selectively deliver therapeutic molecules to phagocytic cells at injured or diseased sites and, at the same time, enabling the visualization of the drug release process. In this context, *Rhodobacter* species also offer promising properties that make them attractive candidates as therapeutic platforms. In contrast to most other Gram-negative bacteria, *Rba. sphaeroides* and *Rba. capsulatus* synthesize non-toxic, non-pyrogenic lipopolysaccharides (LPS) that potently antagonize endotoxin-induced inflammation[41–43]. Thus, *Rhodobacter* cells may not induce deleterious effects per se. Furthermore, *Rhodobacter* can be engineered as gene therapy vectors or to produce secondary metabolites exhibiting anti-cancer activities. For example, the intrinsically high accumulation of pyrrole and isoprenoid scaffolds are a good starting point to utilize this group of photosynthetic bacteria for the production of chlorophylls applicable for photodynamic therapy, or terpenoids such as β-elemene, that have recently been shown to target TAMs in cancer treatment[44–47]. To further extend the applicability of *Rhodobacter* as platform organisms for theranostic research, a diverse set of genetic tools has already been developed allowing easy genetic manipulation. For instance, DNA vectors and integrative cassettes have been constructed for transplanting complete drug pathways into *Rba. capsulatus* and *Rba. sphaeroides*[47–50].

In summary, *Rhodobacter* species offer multiple properties making them promising candidates for preclinical studies of tumor heterogeneity. Our results now demonstrate the usefulness of this class of photosynthetic bacteria by showing that their BChl *a* can be very effectively identified via MSOT and that changes in their unique spectral signature are a function of macrophage activity, which can provide insights into tumor biology in vivo. The broad biosynthetic capability together with the here described reporter properties finally make phototrophic purple bacteria a truly versatile smart micro-packaging system which can help to facilitate the delivery of DNA and active substances into (tumor-associated) macrophages and simultaneously allows MSOT-based in vivo monitoring of drug release in the near future.

## Methods

**Bacterial strains and culture conditions**. All strains used in this study were either grown at anaerobic or microaerobic conditions in different minimal media (RCV[37] with 15 mM ammonia) for *Rba. capulatus*, SISTROM[51] for *Rba. sphaeroides* and CENMED[52,53] for *Rsp. centenum* or in complex media PY[54] for *Rba. capsulatus*. In addition, a mixture of complex PY medium and RCV medium[55] was used for cultivation that was proven to facilitate BChl *a* biosynthesis in *Rba. capsulatus* at 30 °C.

For strain maintenance, the *Rba. capsulatus* wild-type strain B10S (a spontaneous streptomycin-resistant mutant of the original isolate B10)[56], the rifampicin-resistant *Rba. capsulatus* wild-type strain SB1003[57] and the recombinant mutant strain *Rba. capsulatus* Δ*crtJ* (Troost, Loeschcke and Drepper unpublished) were cultivated on 2% (w/v) agar (Bacto agar, Difco) containing PY

plates supplemented with 200 µg ml⁻¹ streptomycin for B10S, 25 µg ml⁻¹ rifampicin for SB1003 or 25 µg ml⁻¹ rifampicin and 10 µg ml⁻¹ spectinomycin for the SB1003/Δ*crtJ* at 30 °C. The *Rba. sphaeroides* wild-type strain ATH 2.4.1[58] and *Rsp. centenum* strain ATCC 43720[52] were cultivated at 30 °C without antibiotics either on PY or CENMED plates containing 2 % (w/v) agar.

Microaerobic (chemotrophic) cultivation of all strains was carried out in Erlenmeyer flasks under permanent shaking (0.57×*g*) in the dark at 37 °C using 50 ml of the above described media without antibiotics. For anaerobic (i.e., phototrophic) cultivation at 37 °C capped air-tight 15-ml reaction tubes were filled with 15 ml of the respective cultivation media without antibiotics to create an oxygen-free atmosphere. For phototrophic growth, different light sources were used emitting either full spectrum light (light bulbs) as well as defined blue ($\lambda_{max}$ = 450 nm) and/or infrared ($\lambda_{max}$ = 850 nm) light. Bulb illumination arrays consist of two panels with three light bulbs (60 W, Osram, Germany) on each panel. LED illumination arrays consist of two LED panels with 120 blue LEDs (NSSB100BT, Nichia, Japan) and 120 high power infrared LEDs (SFH 4257, Osram, Germany) on each panel. Spectral irradiance data are published in ref. [55]. Cell cultures were placed in 13 cm distance to each light panel.

For the analysis of pigment formation, strains were first cultivated in appropriate minimal media (RCV, SISTROM or CENMED) for three days under microaerobic or phototrophic conditions. These pre-cultures were used to inoculate corresponding test cultures in (i) minimal medium (RCV, SISTROM or CENMED), (ii) complex PY medium and (iii) a mixture of complex PY and minimal RCV medium, starting with a cell density of OD₆₆₀ nm = 0.05. The test cultures were cultivated at growth conditions outlined above. After 24 h, cell densities were monitored at 660 nm and BChl *a* accumulation was determined in whole cells by measuring the in vivo absorption at 860 nm spectrometrically (Genesys 6; Thermo Fisher Scientific GmbH, Dreieich, Germany). On the basis of this data, BChl *a* synthesis per cell could be calculated.

For lysed samples, *Rba. capsulatus* was pre-cultivated under anaerobic phototrophic conditions under IR light in RCV medium for 3 days. These cells were used to inoculate ten cultures in hungate tubes with RCV 2/3 PY medium in parallel with a starting cell density of OD₆₆₀ nm = 0.05. After 24 h of cultivation under IR light, cell material corresponding to OD₆₆₀ nm = 400 was pelleted and re-suspended in 20 ml PBS, and subjected to disruption in a French Press apparatus (500 bar). The cell suspension was passed through the apparatus eight times to ensure complete lysis, which was confirmed by plating samples on agar plates (i.e., no colony formation). A 5 µl-sample, corresponding to $1 \times 10^8$ lysed cells, was used for injection.

**Cell culture**. The mouse mammary tumor cell line 4T1 (CRL-2539; American Type Cell Culture Collection, Manassas VA), the mouse colon carcinoma cell line CT26.WT (CRL-2638) and the macrophage line Ana-1 (kind gift from Heiko Adler, Lung Infection & Repair, Helmholtz Zentrum Munich), were grown in 5% CO₂ at 37 °C in RPMI-1640 medium (Sigma-Aldrich) containing 10% fetal bovine serum and 1% antibiotics (penicillin and streptomycin). Routine culture treatment was conducted twice a week. Used cell lines are regularly checked for mycoplasma contamination and authenticity.

**Cell viability assay**. 4T1 cells were inoculated in 96-well plates at $2.5 \times 10^3$ cells per well in a final volume of 100 µl. After overnight incubation for cell adhesion, the medium was removed, and the cells were incubated at 37 °C with 100 µl of medium containing either $3 \times 10^3$ or $3 \times 10^4$ cfu heat inactivated (99 °C or 72 °C for 5 min) bacteria; arguing that a tumor of average weight of 0.3 g contains $3 \times 10^8$ cells with a bacterial colonization of $1 \times 10^8$ cfu this would make 0.3 bacteria per tumor cell; we adjusted the cells in the cell viability assay (Cell Proliferation Kit 1 (MTT), Roche) accordingly. 10% DMSO was used as positive (inducing cell death) and PBS as negative control (no influence on viability). After 24 h of incubation, the medium containing the heat inactivated *Rhodobacter* cells were removed and fresh medium was added. Cell viability was determined by a colorimetric MTT assay used according to the manufacturer's instructions. Absorbance measurements were conducted at 595 nm. For reference absorbance of the untreated PBS control was regarded as 100% and all data normalized accordingly.

**Mouse work**. All animal experiments were approved by the government of Upper Bavaria and were carried out in accordance with the approved guidelines.

For tumor xenografts $0.8 \times 10^6$ 4T1 or CT26.WT cells in phosphate-buffered saline (PBS) have been implanted in the back of Balb c nude mice (Charles River Laboratories, Boston, US) just below the kidney. Mice remained in the animal care facility for several days until the tumors reached a size of 100 mm³ (see Supplementary Figure 2). For MSOT imaging mice have been anaesthetized using 2% Isofluran in O₂. Anaesthetized mice were place in the MSOT holder using ultrasound gel and water as coupling media. The tumor region was imaged from 700 nm to 900 nm with 10 nm step size to cover the spectra of the agent. For bacteria injection 5 µl PBS with $1 \times 10^8$ cfu have been injected from the top directly in the tumor using a 10 µl Hamilton Syringe (gauge 26 s).

After termination of the experiments all mice have been sacrificed and stored at −80 °C for cryosectioning.

**MSOT data acquisition, reconstruction, and unmixing.** All MSOT data have been recorded using the MSOT inVision 256 (iThera Medical GmbH, Kreiling, Germany).

MSOT spectra between 700 and 900 nm of bacterial suspensions have been recorded using clear plastic syringes together with an equally prepared black indian ink reference. For reconstructions, the ViewMSOT software was used with a field-of-view (FOV) of 250 × 250 pixel with a pixel resolution of 200 × 200 μm. Images have been reconstructed using the model linear approach as described in ref. [59] and filtering the data for the range between 0.05 and 7 MHz. Mean MSOT intensities were directly extracted from region of interests (ROI) covering the syringe diameter. For spectral analysis of Ana-1 or 4T1 cells incubated with *Rhodobacter* SB1003 for different timepoints, total mammalian cells were separated from total *Rhodobacter* that has not been endocytosed and recorded using syringes as described above. Mean MSOT intensities for each sample were directly extracted from six different ROI covering the syringe diameter.

To calculate the PGE we measured three concentrations per strain and plotted the OA signal intensities vs. absorption of the scattering corrected absorption spectra at peak wavelength (850 nm for *Rba. spheroides* and 860 nm for *Rba. capsulatus* SB1003 and Δ*crtJ*, respectively). To correct for background signals of the OA measurements the calculated intercept of the linear fit was substrate from each OA signal yielding a linear fit through zero and through all data points. The calculated slope of this fit represents the conversion of absorbed photons to OA signal for the given solution. We referenced all values to a reference slope with 2.6, 5.2, and 7.8 μM solutions of methylene blue.

Mouse MSOT data between 680 and 960 nm have been recorded in 13 (early tumor) to 34 slices (late tumor development) with 0.5 mm spacing covering the tumor region. Details of MSOT recording are described elsewhere[60]. The field-of-view (FOV) was 200 × 200 pixel with a pixel resolution of 100 × 100 μm. Images have been reconstructed using a model-based approach as described in[59] and filtering the data for the range between 0.1 and 7 MHz. Speed of sound was regularly set to 1530 m/s and a regularization parameter was used with $10^6$. A ROI covering the mouse body was selected for overall data analysis (body-ROI), as well as an ROI for specific analysis of signals from the tumor region (tumor-ROI). To obtain the accumulative spectra over the tumor the intensities of all pixels in the tumor-ROI over all slices were summed for each wavelength.

Unmixing for the range of 700 to 900 nm was performed essentially as described in ref. [61]. Target spectra used for unmixing are indicated with the data. For the accumulative quantification of the unmixed OA signal OA intensities of the unmixed images are summed over all positively unmixed pixels in the tumor-ROI.

For the spectral analysis the spectra at each pixel have been analyzed for their peaks using Matlab. For spectra where the 800 nm peak only occurs as a shoulder the contribution was assessed by using the ratio of the main peak at ~860 nm to the spectral position where the 800 nm peak can be assumed (i.e., 800 nm). Unmixed pixel where automated peak assignment failed have been omitted.

**Cryosectioning with fluorescence imaging.** After sacrificing by cervical dislocation, the mice were cryopreserved at −80 °C. To detect BChl *a*-mediated fluorescence in tumors as shown in Supplementary Figure 12, 250 μm sections were cut (Leica CM1950, Leica Microsystems, Wetzlar Germany) and imaged using a 740/40 nm bandpass for excitation and 850 nm longpass filter for detection. Images were taken using an Andor LucaR CCD camera (DL-604M, Andor Technology, Belfast, UK) with 10 s exposure and a gain of 40.

**Immunofluorescence, histochemistry, and microscopy.** For immunofluorescence (IF), Ana-1 macrophages were grown on poly-L-lysine coated coverslips following addition of *Rhodobacter* SB1003 at a ratio of 50:1 (Ana-1:Rba) for 24 h. Coverslips were briefly washed with pre-warmed PBS, fixed for 7 min in 4% pre-warmed paraformaldehyde (PFA) followed by IF staining as described earlier[62]. The primary polyclonal rabbit antibody to Lamp-1 (Lysosome marker, Abcam, #24170, diluted 1:500) was paired with the secondary antibody Alexa 405 anti-rabbit.

For immunohistochemistry (IHC) staining of frozen tissue sections (10 μm) the protocol was altered the following: Sections stored at −80 °C were transferred to room temperature, equilibrated for 30 min and fixed for 10 min with Acetone. The primary rat monoclonal antibody recognizing the mouse F4/80 antigen (Macrophage-specific marker, Abcam, #6640, 1:500) and the polyclonal rabbit antibody against Cleaved Caspase 3 (Merck Millipore, AB3623, 1:300) were combined with anti-rat and anti-rabbit peroxidase/HRP conjugated antibodies at standard dilutions, respectively, followed by DAB detection (Roth 9202.1). H&E staining was performed according to standard protocol.

Microscopic images showing IF with anti-Lamp-1 were collected with the Zeiss Axio Imager M2 using the AxioCam MRm camera and the Plan Apochromat 100×/1.4 oil objective to record Z-stacks with 250 nm step size. The Zeiss Zen software was used for image processing and deconvolution at default (constrained iterative method) was applied. Images shown in Fig. 2f and Supplementary Figure 11 are grouped Z-projections of maximum intensity. Brightfield IHC images were recorded with the AxioColor camera and the EC Plan Neofluar 2.5×/0.085 and 40×/0.75 Ph2 objectives. Tiling of the entire tumor was executed using the 10×/0.3 Ph1 objective, the AxioCam MRm camera and filter settings for Dapi and Cy7. The following filtersets have been used for fixated samples (Zeiss): DAPI

(FilterNr. 49) Ex: 335–383 nm Em: 420–470 nm, HE Red (FilterNr. 63 = mCherry): Ex: 559–585 nm Em: 600–690 nm and Cy7: Ex: 673–748 nm Em: 765–855 nm. While life cultures have been imaged with (Leica): HQCherry: Ex: 580/20 nm Em: 630/60 nm, Y7: Ex: 710/75 Em: 810/90 nm.

## Data availability

Due to size limitations of depositing raw imaging data the data that support the findings of this study are available from the corresponding author upon request.

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

## Acknowledgements

A.C.S. wishes to thank R. Hillermann for technical assistance, I. Olefir for technical discussions, and A.C. Rodríguez and A. Heck for discussions on the manuscript. The work of L.P. and A.C.S. was supported by the Deutsche Forschungsgemeinschaft (DR 785/1-1 and STI 656/1-1, respectively).

## Author contributions

L.P. and A.L. performed the molecular and microbiological experiments, analyzed the data and prepared the bacteria. I.W. designed and performed the mammalian cell experiments as well as histology, analyzed the data, and contributed to the manuscript. U.K. performed mouse work and MSOT measurements. R.W. helped with the bacterial work. K.E.J. contributed to the manuscript. T.D. designed the experiments, analyzed the bacterial data, and wrote the manuscript. V.N. conceived the OA measurement setup and contributed to the manuscript. A.C.S. conceived and designed the experiments, analyzed the MSOT data, and wrote the manuscript.

## Additional information

**Competing interests:** V.N. is a shareholder of iThera Medical GmbH. The remaining authors declare no competing interests.

9