## [Peer Review File · Nature Communications]

Reviewers' comments:

Reviewer #1 (Remarks to the Author):

This manuscript, "The phototrophic bacterium Rhodobacter as an optoacoustic in vivo reporter of tumor heterogeneity and dynamics", introduced Rhodovacter as a reporter to visualize heterogeneities of tumor using optoacoustic imaging using difference in optical absorption of the bacterial system depending tumor microenvironment such as oxygen concentration. The authors provided sufficient in vivo optoacoustic (OA) spectral analysis results in various type of bacteria strains and tumor model. As a result, Rba. Capsulatus SB1003 cells clearly showed the spectral change after injection into the 4T1 tumor which indicates the Rba. Capsulatus SB1003 can be used as an OA biosensor of tumor. This paper can be accepted after the following comments are addressed.

Comments

1. In figure S5, S7, S8 and 1B, the OA signal seems to be normalized at every time point. Did the OA amplitude decreased over time? If the amplitude did not decrease, did the bacterial injected into the tumor moved to other organs or out of the body?
2. In figure 2 and S5, the spectral change (red shift) of CT26.WT model is less then 4T1 model. What can be the reason for these results?
3. The longer the time passes, the less the OA signal will be, unless the material is injected and not spreading. If the OA signal is reduced, the accuracy of the spectral analysis is likely to decline. Do you think the accuracy is high enough to be used as a biosensors of tumor biology?
4. What is the exact mechanism of the spectral change of injected bacteria in 4T1 tumor? Is there any spectral change overtime if the bacteria is injected in normal tissue?
5. It seems to be better if the spectral change of CT26.WT model is also added in figure 2.
6. The statistics for the animal work should be described in the figure 2 caption.

Reviewer #2 (Remarks to the Author):

This work applied BChl a of purple bacteria together with optoacoustic imaging to study the tumor spatiotemporal heterogeneity. The results showed that the changes of the optoacoustic spectral signature of BChl a can be used to reveal the individual tumor microenvironment, showing BChl a containing bacteria may be good for preclinical studies of tumor heterogeneity and assessment of tumor intervention. Overall, the work presents some good novelty, however, the potential applications need to be further illustrated and demonstrated. The following are my comments:

1. The variation of the bacteria optoacoustic signal in vivo in this study is assumed to be influenced by oxygen level in the tumor. What about other factors of tumor microenvironment such as pH and extracellular molecular composition as indicated in the Introduction part. Even though oxygen is the main factor, the authors should provide data with other means (better can be regarded as golden standard) for cross validating of this.
2. Although the manuscript is mainly focused on preclinical application of the bacteria, the authors are suggested to provide more discussions on its potential for clinical translation.

3. In the Discussion part, the authors concluded that the bacteria is promising to be ultimately used for tumor diagnosis. However, for the in vivo study in this paper, the bacteria is directly injected to the tumor site, which assumes the location of the tumor has been accurately determined. The authors should discuss how the bacteria can be used in real condition for tumor diagnosis.

4. The in vivo optical absorbance intensity of the bacteria ultimately determines its optoacoustic performance. The authors should compare the absorbance intensity of the bacteria in vivo to that of blood, and provide the relevant data.

5. Only in vitro toxicity of the bacteria was evaluated in this study. The authors should provide in vivo biosafety data.

6. The stability of BChl a of purple bacteria under laser excitation should be evaluated.

7. BChl a was generated in three specific bacteria as shown in this manuscript. This may limit its applications to a relatively narrow area. Please comment and elaborate on the potential applications in a wider area.

Reviewer #3 (Remarks to the Author):

This study aims at using the opto-acoustic signals generated by two separate absorption bands in the optical absorption of the LH1 and LH2 antennas of photosynthetic bacteria as reporters for the microenvironment heterogeneity of the imaged tumors. Two animal models have been used to test this concept-4T1 mammary tumors and CT26 colon carcinoma tumors both grafted SC in the back of Balb C mice.

General comments:

The heterogeneity of the tumor microenvironment has become a major research objective for both understanding cancer progression and development therapeutic approaches. Currently, single cell analyses, multiplex staining etc can provide fairly detailed information using dissected domains from tumor specimen and different types of biopsies. However, there is an urgent need for in vivo monitoring for better diagnoses, prognosis and new therapeutics design. Using the spectral properties of non-pathogenic bacteria as tissue reporter offered by the authors of the currently evaluated manuscript is a nice concept although novelty should be granted to previous authors some of whom (e.g. Cronin et al) are cited in the manuscript. Notably, using the optical absorption as generator of SMOST signals is original and should provide better resolution than fluorescence or bio-luminescence monitoring. Unfortunately, the meaning of the currently presented results is fairly unclear. Moreover, I am not convinced that in the current state of the art, the proposed technology offers any advantageous on other imaging modalities for the purpose of monitoring the tumor microenvironment. Hence, I do not recommend publication of this study.

This recommendation does not rule out different opinion in case the authors clearly connect the collected data with meaningful mechanism of action and biological assignments based for example on histology, immunohistochemistry and other approaches that validate their overall conclusions.

Specific comments

1. The spectral changes in Fig. 2, shifting with time from double peak spectra (800/850-60nm) to a single peak at 850 nm, may either reflect bleaching of the LH2 antenna while maintaining the LH1 or, bleaching of the non-coupled Bchl a in the LH1 antenna, or simply and most likely, lysis of the cells, disintegration of the Bchls from their protein scaffolds and re-aggregation as non-protein bound or protein bound molecules to large structures that absorb at 850-60 nm. Such events have been carefully documented and reported in the extensive literature dealing with Bchl aggregation. This process can be enhanced by macrophage infiltration to the necrotic tumor domains. To avoid misinterpretation the authors can perform micro-dissection and single cell analysis. Alternatively, they should look at the "free" Bchl using literature approaches. Please note that no significant increase in the volume occupied by the 800/850 nm signals can be observed casting shade on the

possible growth of a different form of the in situ bacteria.

2. The increased contribution of "blood" signals (white) at the central tumor domain is confusing since this is the necrotic domain. It may therefore be due to hemoglobin (oxy/deoxy) changes following disintegration of the vascular system near the necrotic domain.

3. A major deficiency of the paper is the lack of histological support. This is quite disappointing because the author could use the intrinsic absorption and fluorescence of the Bchl_s to look at cellular integrity, allocate the cells and informed their location with the tumor microenvironment. Moreover, it would be highly informative to check for immune infiltration, inflammation etc.

Comments concerning the presentation:

There is a substantial lack of information in the figure legends, tables etc. Moreover, some legends do not refer to the presented data. A few examples:

1. Figure 1:A: For each strain , data are shown for tree mice before injection (green), immediately after injection (red) and at...." This legend should go with B, which shows changes in the bacterium optical absorption. Furthermore, the grey lines should refer to tim intervals post injection.

2. Figure 7: the title does not match and/or reflect the presented data but rather treatment conditions.

3. There are many other examples.

**For reasons of discussion we change the order of the reviewers' comments and begin with addressing the comments of reviewer #3.
Please note that we provide the file "Peters_et_al_RESUB_manuscript_MARKUP.doc" with the major changes marked in yellow.**

Reviewer #3 (Remarks to the Author):

This study aims at using the opto-acoustic signals generated by two separate absorption bands in the optical absorption of the LH1 and LH2 antennas of photosynthetic bacteria as reporters for the microenvironment heterogeneity of the imaged tumors. Two animal models have been used to test this concept-4T1 mammary tumors and CT26 colon carcinoma tumors both grafted SC in the back of Balb C mice.

General comments:

The heterogeneity of the tumor microenvironment has become a major research objective for both understanding cancer progression and development therapeutic approaches. Currently, single cell analyses, multiplex staining etc can provide fairly detailed information using dissected domains from tumor specimen and different types of biopsies. However, there is an urgent need for in vivo monitoring for better diagnoses, prognosis and new therapeutics design. Using the spectral properties of non-pathogenic bacteria as tissue reporter offered by the authors of the currently evaluated manuscript is a nice concept although novelty should be granted to previous authors some of whom (e.g. Cronin et al) are cited in the manuscript. Notably, using the optical absorption as generator of SMOST signals is original and should provide better resolution than fluorescence or bioluminescence monitoring. Unfortunately, the meaning of the currently presented results is fairly unclear. Moreover, I am not convinced that in the current state of the art, the proposed technology offers any advantageous on other imaging modalities for the purpose of monitoring the tumor microenvironment. Hence, I do not recommend publication of this study.

This recommendation does not rule out different opinion in case the authors clearly connect the collected data with meaningful mechanism of action and biological assignments based for example on histology, immunohistochemistry and other approaches that validate their overall conclusions.
Specific comments

1. The spectral changes in Fig. 2, shifting with time from double peak spectra (800/850-60nm) to a single peak at 850 nm, may either reflect bleaching of the LH2 antenna while maintaining the LH1 or, bleaching of the non-coupled Bchl_a in the LH1 antenna, or simply and most likely, lysis of the cells, disintegration of the Bchls from their protein scaffolds and re-aggregation as non-protein bound or protein bound molecules to large structures that absorb at 850-60 nm. Such events have been carefully documented and reported in the extensive literature dealing with Bchl aggregation. This process can be enhanced by macrophage infiltration to the necrotic tumor domains. To avoid misinterpretation the authors can perform micro-dissection and single cell analysis. Alternatively, they should look at the "free" Bchl using literature approaches. Please note that no significant increase in the volume occupied by the 800/850 nm signals can be observed casting shade on the possible growth of a different form of the in situ bacteria.

We are very grateful that the reviewer pointed out this perspective to us. In our initial work we considered disintegrated bacteria and tested their spectra by analyzing lysed bacteria through heat-inactivation or french-press, assuming such states of disintegration to resemble the situation after ingestion of bacteria by e.g. macrophages (these results are now also added as Supplementary Figure 10). In response to the reviewers' comments we now additionally conducted a study in which we observed the spectral changes when Rhodobacter cells are co-cultivated with 4T1 tumor cells and Ana-1 macrophages in vitro. Indeed, in case of the macrophages but not the tumor cells we could demonstrate ingestion of Rhodobacter and a spectral change that resembles the situation we observe in the tumor. This confirms the reviewer's suggestion that the change in spectral signature is not based on a changed metabolism but in contrast indicates the specific activity of macrophages. Based on this new observation we completely revised the manuscript now following the activity of macrophages in vivo which shows dependence of tumor models as well as bacterial strains and growth conditions. This gives us a live insight into macrophage activity with high spatial and temporal resolution; thus this pilot study formulates the basis for a multitude of applications to research macrophage behavior and their role in diverse diseases including cancer. Again, we are very indebted to reviewer #3 for redirecting our study in this new and very exciting direction. In the revised manuscript we added Figure 2 showing above mentioned in vitro experiments and added a paragraph termed "Analysis of the changed spectral signature" along with Supplementary Figures 9 to 14.

2. The increased contribution of "blood" signals (white) at the central tumor domain is confusing since this is the necrotic domain. It may therefore be due to hemoglobin (oxy/deoxy) changes following disintegration of the vascular system near the necrotic domain.

The white signals in the images at 720 nm point to a strong signal at that wavelength without a clear assignment for Rhodobacter (800/850 nm or 850 nm alone). Since this is frequently seen at the center of

the tumor at later timepoints this can be likely attributed to deoxygenated blood. We note that this partially obscure presence of bacterial material in the same area. We however also note that the shifted spectra (850 nm alone) are regularly stronger – making detection of macrophages action feasible also on a relatively strong background (also see Reviewer Comments Figure 1).

Figure 1: MSOT images unmixed for *Rba.* and color coded for spectral content (A) and at 720 nm (B). Spectral data for the three positions indicated in the images (C).

3. A major deficiency of the paper is the lack of histological support. This is quite disappointing because the author could use the intrinsic absorption and fluorescence of the BChls to look at cellular integrity, allocate the cells and informed their location with the tumor microenvironment. Moreover, it would be highly informative to check for immune infiltration, inflammation etc.

We added now extensive histological studies to the manuscript (Supplementary Figure 13). We stained for presence of macrophages (F4/80) as well as cell-death (Caspase-3) and used a Hematoxylin and eosin staining to inspect the general cellular integrity. Additionally, we probed the fluorescence of BChl a. In line with our (new) rationale we see macrophages aggregation in the zone of BChl a fluorescence together with strong cell-death at the center. Moreover, we added immune staining and fluorescence microscopy data to identify the intracellular fate of the ingested Rhodobacter (Supplementary Figure 11 and 12). Interestingly, lysosomal marker Lamp-1 and fluorescence from ingested Rhodobacter / BChl a do not co-localize suggesting a stalled phagosomal pathway.

Comments concerning the presentation:

There is a substantial lack of information in the figure legends, tables etc. Moreover, some legends do not refer to the presented data. A few examples:

1. Figure 1:A: For each strain , data are shown for tree mice before injection (green), immediately after injection (red) and at....” This legend should go with B, which shows changes in the bacterium optical absorption. Furthermore, the grey lines should refer to tim intervals post injection.

The quoted figure legend is correct, since A describes the absolute signal at different time-points. The spectral data is given in Figure 1B.

2. Figure 7: the title does not match and/or reflect the presented data but rather treatment conditions.

This figure is now included in the histology analysis figure (Supplementary Figure 13).

3. There are many other examples.

However, we take the reviewers comment and improved readability and clarity throughout the manuscript. Moreover, we deleted several Supplementary Figures of the original manuscript to increase readability.

Reviewer #1 (Remarks to the Author):

This manuscript, “The phototrophic bacterium Rhodobacter as an optoacoustic in vivo reporter of tumor heterogeneity and dynamics”, introduced Rhodobacter as a reporter to visualize heterogeneities of tumor using optoacoustic imaging using difference in optical absorption of the bacterial system depending tumor microenvironment such as oxygen concentration. The authors provided sufficient in vivo optoacoustic (OA) spectral analysis results in various type of bacteria strains and tumor model. As a result, *Rba. Capsulatus* SB1003 cells clearly showed the spectral change after injection into the 4T1 tumor which indicates the *Rba. Capsulatus* SB1003 can be used as an OA biosensor of tumor. This paper can be accepted after the following comments are

addressed.

Comments

1. In figure S5, S7, S8 and 1B, the OA signal seems to be normalized at every time point. Did the OA amplitude decreased over time? If the amplitude did not decrease, did the bacterial injected into the tumor moved to other organs or out of the body?

All spectra if not state otherwise (e.g. normalized to 850) are mean OA amplitudes of the ROI or unmixed pixels. In regard to the reviewer's question on the "decreasing" amplitude. Since new data now suggests that the signal, especially at later timepoints, mainly stems from BChl a ingested by macrophages we can assume that the chromophore get eventually catabolized resulting in a decaying signal. In contrast, a strong proliferation of Rhodobacter could not be observed within our time window of observation.

2. In figure 2 and S5, the spectral change (red shift) of CT26.WT model is less then 4T1 model. What can be the reason for these results?

This is one of the interesting observations that we made and now discuss in the manuscripts. The lesser red shift points to a lower activity of macrophages in this area and tumor model. It is tempting to assume that this reflects a difference in number or activity of tumor associated macrophages in CT26.WT and 4T1, in line with our proposition to use the spectral shift of Rhodobacter upon macrophage ingestion as a marker for the latter.

3. The longer the time passes, the less the OA signal will be, unless the material is injected and not spreading. If the OA signal is reduced, the accuracy of the spectral analysis is likely to decline. Do you think the accuracy is high enough to be used as a biosensors of tumor biology?

Since we now could demonstrate that the relevant spectral-shift is most probably due to an irreversible ingestion of Rhodobacter by macrophages the observation time is intrinsically limited. For our experiments lasting up to 8 days signal intensity is sufficiently strong to monitor the spectral-changes. Moreover, the extends of the signal seem to be more or less constant suggesting that all regions of the tumor receiving bacteria during injection have sufficient signal to sustain at least an 8 day measurement.

4. What is the exact mechanism of the spectral change of injected bacteria in 4T1 tumor? Is there any spectral change overtime if the bacteria is injected in normal tissue?

As outlined above (see response to comments of reviewer #3) we now could identify the ingestion of Rba. capsulatus cells and subsequent disintegration by macrophages followed by aggregation of monomeric BChl a as the reason for the spectral shift. We added Figure 2 and paragraph "Analysis of the changed spectral signature" to explain this phenomenon. As suggested by the reviewer, we performed an additional in vivo experiment where Rhodobacter cells were also injected into the muscle and added Figure 3 panel C in the manuscript. In line with high vascularization and macrophages presence in the muscle we see a fast and changes of the spectral signature.

5. It seems to be better if the spectral change of CT26.WT model is also added in figure 2.

The spectra in Figure 3 (former 2) are based on an average of the pixels showing a given ratio of 800/860 nm peaks. In that sense they can be seen as exemplary spectra and apply for all panels shown in figure 3. We reprocessed all data now including our new in vivo data so that the same scaling applies throughout the study.

6. The statistics for the animal work should be described in the figure 2 caption.

We thank the reviewer for that comment and added more clear information about the animals used in each group to the manuscript (Supplementary Table 1).

Reviewer #2 (Remarks to the Author):

This work applied BChl a of purple bacteria together with optoacoustic imaging to study the tumor spatiotemporal heterogeneity. The results showed that the changes of the optoacoustic spectral signature of BChl a can be used to reveal the individual tumor microenvironment, showing BChl a containing bacteria may be good for preclinical studies of tumor heterogeneity and assessment of tumor intervention. Overall, the work presents some good novelty, however, the potential applications need to be further illustrated and demonstrated. The following are my comments:

1. The variation of the bacteria optoacoustic signal in vivo in this study is assumed to be influenced by oxygen level in the tumor. What about other factors of tumor microenvironment such as pH and extracellular molecular composition as indicated in the Introduction part. Even though oxygen is the main factor, the authors should provide data with other means (better can be regarded as golden standard) for cross validating of this.

During revision we identified ingestion and disintegration by macrophages followed by aggregation of the BChl a as the reason for the spectral shift. We added Figure 2 and paragraph "Analysis of the changed spectral signature" to explain this phenomenon. This new data explains the spectral-shift convincingly and

we thus retract our prior assumption of the shift being linked to a change growth-conditions of live Rhodobacter.

2. Although the manuscript is mainly focused on preclinical application of the bacteria, the authors are suggested to provide more discussions on its potential for clinical translation.

We clearly see our work in the context of pre-clinical research in tumor, immunology or macrophages biology as well as elucidating effects of pathogen defense (explained in the discussion). Although an application to e.g. probe the tumor associated macrophage population in human could be envisioned the system clearly needs further research and understanding before applications for clinical use can be extracted.

3. In the Discussion part, the authors concluded that the bacteria is promising to be ultimately used for tumor diagnosis. However, for the in vivo study in this paper, the bacteria is directly injected to the tumor site, which assumes the location of the tumor has been accurately determined. The authors should discuss how the bacteria can be used in real condition for tumor diagnosis.

Since an accumulation of i.v. injected Rhodobacter could be demonstrated by Kwon et al. 2014 it is conceivable that the system and the changes could be eventually also used in a diagnostic context. In our present work we however focus on the direct injection of the bacteria in the tumor environment towards actively analyzing tumor associated macrophage activity. Further, research into Rhodobacter as a platform organism can clearly focus on merging strategies, e.g. targeting, drug release and spectral change indicating macrophage activity.

4. The in vivo optical absorbance intensity of the bacteria ultimately determines its optoacoustic performance. The authors should compare the absorbance intensity of the bacteria in vivo to that of blood, and provide the relevant data.

We thank the reviewer for that comment and added a comparison of Rhodobacter against blood (Manuscript page 3 bottom paragraph).

5. Only in vitro toxicity of the bacteria was evaluated in this study. The authors should provide in vivo biosafety data.

We could observe no systemic effect on the animal in our observation window as judged by animal welfare and weight. This is corroborated by results from Kwon et al. that detected no systemic infection or inflammation using C-reactive protein and Procalcitonin assays.

6. The stability of BChl a of purple bacteria under laser excitation should be evaluated.

As suggested by the reviewer, we added a photo-fatigue experiment utilizing live Rhodobacter (Supplementary Figure 6).

7. BChl a was generated in three specific bacteria as shown in this manuscript. This may limit its applications to a relatively narrow area. Please comment and elaborate on the potential applications in a wider area.

Translation of BChl a production and stabilization are clearly longstanding and demanding tasks that go far beyond this study. However, we think that especially in context with our new perspective on macrophages, the amiability of Rhodobacter to genetic manipulations make Rhodobacter a promising bacterial agent that might allow a selective delivery of therapeutic molecules to phagocytic cells at diseased sites and, at the same time, enabling the visualization of the drug release process in near future.

REVIEWERS' COMMENTS:

Reviewer #1 (Remarks to the Author):

The authors well replied to my previous comments. Now, it is ready to be published.

Reviewer #2 (Remarks to the Author):

No further questions. The manuscript may be accepted to be published in Nature Communications.